# Mandatory Notification of Panton–Valentine Leukocidin-Positive Methicillin-Resistant *Staphylococcus aureus* in Saxony, Germany: Analysis of Cases from the City of Leipzig in 2019

**DOI:** 10.3390/microorganisms11061437

**Published:** 2023-05-29

**Authors:** Utta Helbig, Constance Riemschneider, Guido Werner, Nancy Kriebel, Franziska Layer-Nicolaou

**Affiliations:** 1Department of Hygiene, Local Health Authority City of Leipzig, Rohrteichstraße 16-20, 04347 Leipzig, Germany; utta.helbig@leipzig.de (U.H.); constance.riemschneider@leipzig.de (C.R.); 2National Reference Centre for Staphylococci and Enterococci, Division of Nosocomial Pathogens and Antimicrobial Resistances, Department of Infectious Diseases, Robert Koch-Institute, Wernigerode Branch, Burgstraße 37, 38855 Wernigerode, Germany; wernerg@rki.de (G.W.); kriebeln@rki.de (N.K.)

**Keywords:** *Staphylococcus aureus*, Panton–Valentine Leukocidin, notification, community

## Abstract

In Germany, Saxony is the only federal state where the detection of a Panton–Valentine Leukocidin (PVL)-positive Methicillin-resistant *Staphylococcus aureus* (MRSA) has to be notified to the local health authority (LHA). The LHA reports the case to the state health authority and introduces concrete infection control measures. We analyzed isolates from the respective cases in 2019, which were collected in local microbiological laboratories and sent to the National Reference Centre (NRC) for Staphylococci and Enterococci for strain characterization and typing. Antibiotic resistance testing was done by broth microdilution. Molecular characterization was performed using *spa* and SCC*mec* typing, MLST, and the PCR detection of marker genes associated with distinct MRSA lineages. Demographic and clinical data of the individual cases were assessed and the LHA performed epidemiological investigations. Thirty-nine (index) persons, diagnosed with a PVL-positive MRSA, were initially reported to the LHA. Most patients suffered from skin and soft-tissue infections (SSTI). For 21 of the index cases, household contacts were screened for MRSA. Seventeen out of 62 contacts were also colonized with a PVL-positive MRSA. The median age of altogether 58 individuals was 23.5 years. In over half of the cases, the home country was not Germany and/or a history of travel or migration was reported. Molecular characterization revealed the presence of various epidemic community-associated MRSA lineages, with “USA300”, including the North American Epidemic (ST8-MRSA-IVa) and the South American Epidemic Clone (ST8-MRSA-IVc), the “Sri Lankan Clone” (ST5-MRSA-IVc), and the “Bengal Bay Clone” (ST772-MRSA-V) being more prevalent. In eight out of nine households, the contact persons were colonized with the same clone as the respective index case, suggesting a close epidemic and microbiological link. The obligation to report PVL-positive MRSA enables us to detect the occurrence of PVL-producing MRSA and its spread in the population as early as possible. Timely detection allows the targeted deployment of reliable anti-infective measures.

## 1. Introduction

For decades, Methicillin-resistant *Staphylococcus aureus* (MRSA) has been one of the most commonly detected pathogens of nosocomial infections. In addition to hospital-acquired MRSA (HA-MRSA) and hospital-acquired community onset (HCO) MRSA, which are strains in the community that are often epidemic strains acquired during hospitalization and persist for a long time as colonizers, MRSA are also detected as nasal colonizers and causative agents of infections in the non-hospitalized population (community-acquired MRSA, CA-MRSA) and in humans having close contact with livestock (livestock-associated MRSA, LA-MRSA) [1,2].

CA-MRSA have been described since the late 1990s, mainly in the outpatient sector, and are associated with skin and soft-tissue infections and also severe systemic infections such as necrotizing pneumonia in otherwise healthy individuals. Many CA-MRSA strains are carrying genes encoding the bi-component toxin Panton–Valentine Leukocidin (PVL), which appears to be a virulence factor associated with necrotic lesions of the skin and subcutaneous tissues and also with community-acquired severe necrotic disease patterns [3,4,5]. Compared to HA-MRSA, CA-MRSA are characterized by different clonal backgrounds, smaller SCC*mec* cassettes coding for Methicillin resistance and plasmids with additional resistance genes [2,6].

The burden of disease caused by CA-MRSA varies widely around the world, whereas systematic data and studies for many countries and regions are limited [1,7,8,9]. In general, the epidemiology of MRSA is characterized by the serial emergence of predominant strains, having the ability to spread widely. Since the 1990s, genotypic differences by site of acquisition have begun to homogenize, demonstrating that CA-MRSA and HA-MRSA can each invade the other’s niche [2].

In Germany, the legal basis for the control of infectious diseases is provided by the Protection against Infection Act. In addition, there are laws and ordinances in the individual federal states that extend the reporting obligations under the Infection Protection Act. Of all federal states in Germany, only in Saxony does the detection of a PVL-positive MRSA have to be notified by name from the laboratory to the local health authority (LHA), when evidence indicates acute infection. The LHA reports the case to the state health authority and has to take concrete infection control measures after assessing each individual case.

In the following study, we analyzed in detail PVL-positive MRSA and the respective cases from the city of Leipzig, which were reported to the LHA as part of the mandatory reporting in Saxony in the year 2019. Our study shows how a surveillance system for PVL-positive MRSA contributes to valuable data for molecular surveillance of the pathogen and enables early diagnosis and treatment. It should raise awareness to the related disease manifestations and targeted diagnostic evaluation, especially among physicians in primary healthcare settings, in order to improve patient care.

## 2. Material and Methods

### 2.1. Laboratory Sampling and Data Collection

Since 2012, there is an obligation in Saxony to report the detection of PVL-positive, community-acquired MRSA, when evidence indicates acute infection [10]. A targeted microbiological diagnosis for PVL-positive MRSA in the outpatient setting or at hospital admission was carried out by local laboratories in the following cases: recurrent and/or deep-seated skin and soft-tissue infection; a previous stay in areas with high CA-MRSA prevalence; close contact to patients with PVL-positive MRSA.

The cases were subsequently reported by the local laboratories to the local health authority (LHA) City of Leipzig. The LHA contacted the respective patients by mail and invited them to a structured interview (for reporting and contact form, see Appendix A) to obtain further data for reporting the case to the state health authority and to introduce concrete infection control measures. These measures included, among others, the identification of contact persons and comprehensive information on MRSA and MRSA decolonization. To prevent recurrent infection, MRSA decolonization was performed, consisting of 5 days of topical therapy of the nasal mucosa, antiseptic skin cleansing, and throat rinsing [11]. Furthermore, the clinical isolates were sent to the NRC on request of the LHA for further characterization. Only the first isolate per patient was included.

### 2.2. Identification and Antimicrobial Susceptibility Testing

At the NRC, the isolates were cultured on sheep blood agar and *S. aureus* species were confirmed by the presence of the clumping factor and by the tube coagulase test using human plasma. Antimicrobial susceptibility of the isolates was determined by broth microdilution according to EUCAST (www.eucast.org (accessed on 1 January 2019)). The following antibiotics were tested: penicillin (BEN), oxacillin (OXA), cefoxitin (CXI), fosfomycin (FOS), gentamicin (GEN), linezolid (LIN), erythromycin (ERY), clindamycin (CLI), tetracycline (TET), tigecycline (TIG), vancomycin (VAN), teicoplanin (TEI), ciprofloxacin (CIP), mupirocin (MUP), moxifloxacin (MOX), daptomycin (DAP), fusidic acid (FUS), rifampicin (RIF), trimethoprim/sulfamethoxazol (TRS), oxacillin/sulbactam (OxaSu).

### 2.3. DNA Extraction

Strains were grown overnight in tryptic soy broth at 37 °C. DNA was extracted using the DNeasy tissue kit (Qiagen, Hilden, Germany) according to the manufacturer’s instructions with the modification that lysostaphin (100 mg/L; Sigma, Munich, Germany) was added to the cell-lysis step.

### 2.4. Detection of Resistance and Virulence Genes

In addition to the antimicrobial susceptibility testing, MRSA were confirmed by the PCR-detection of the *mecA* gene. The protocol was applied as published by Murakami et al. [12], but using an alternative forward primer (5′-TGG CTC AGG TAC TGC TAT CCA C-3′). A multiplex PCR was performed to detect markers associated with CA-MRSA, including the Panton–Valentine Leukocidin gene (*lukPV*), the enterotoxin H gene (*seh*) for CA-MRSA of clonal lineage ST1/USA400, the arginine deiminase gene (*arcA*) as part of the ACME (arginine catabolic mobile element) cluster for ST8/t008/USA300, and the gene for exfoliative toxin D (*etd*) of clonal lineage ST80 [13].

To assign strains 19-00646, 19-00928, 19-01169, 19-01274, 19-01359, 19-02721, 19-03002, and 19-03580 to the South American Epidemic (SAE), USA300 lineage genomic DNA was sequenced on a NextSeq platform in paired-end mode with a final readout of 2 × 250 bp. The quality of raw read data was assessed by an in-house-developed pipeline (QCumber-2) and reads were assembled using SPAdes software version 3.11.1 [14]. Geneious Prime version 2021.2.2 was used to detected coding sequences of the copper and mercury resistance factor (COMER) region [15] using MRSA CA12 as a USA300 SAE reference genome (accession number CP007672). Genome raw data of 19-00646, 19-00928, 19-01169, 19-01274, 19-01359, 19-02721, 19-03002, and 19-03580 were deposited to the Sequence Read Archives (SRA) of the NCBI (accession numbers: available at release on 31.05.23) under BioProject number PRJNA957965.

### 2.5. Molecular Typing

SCC*mec* elements were classified as described elsewhere [16,17]. For subtyping of SCC*mec* IV cassettes, a Multiplex PCR by Milheirico et al. was applied [18]. For *spa* typing, the amplification and sequencing of the polymorphic X-region of the protein A gene (*spa*) was performed according to [19,20]. *Spa* types were grouped into *spa* clonal complexes (*spa* CC) using the BURP (Based Upon Repeat Pattern) algorithm of the Ridom Staphtype Software (version 2.2.1, Ridom GmbH, Würzburg, Germany). Multilocus sequence typing (MLST) on *S. aureus* was done as published by Enright et al. [21]. Allelic profiles and sequence types were assigned using the MLST database for *S. aureus* (http://saureus.mlst.net/ (accessed on 14 December 2020)).

### 2.6. Definitions

Index persons were patients with diagnosed PVL-positive MRSA reported to the LHA. Household contacts were defined as persons living in the same house as the initial index person or having frequent contact in the same house with the index person.

### 2.7. Confidentiality and Ethical Statement

Data and procedures were part of the mandatory reporting of PVL-positive MRSA in Saxony as authorized by the German Protection against Infection Act. All methods were performed in accordance with the relevant guidelines and regulations. Within the respective institutions, medical confidentiality and data protection were fully guaranteed.

## 3. Results

### 3.1. Bacterial Isolates and Patient Population

Fifty-eight MRSA isolates were obtained from 30 individual cases and from 28 cases related to 2–7 other isolates (mostly identified household contacts). The persons were between one and 79 years old (median age 23.5 years; 44.8% female, 55.2% male). In 10 cases, the place of residence was outside the district of Leipzig. Demographic and clinical data of all individuals are summarized in Appendix A.

Thirty-two MRSA were isolated from skin and soft-tissue infections (SSTI) at different body sites. Three strains originated from MRSA-infections other than SSTI. Twenty-one isolates were obtained from MRSA screening at hospital admission or resulted from screening of household contacts. The origin of strain by patient age is summarized in Figure 1.

In 19 (32.8%) cases, the home country was not Germany, but Afghanistan, Georgia, India, Iran, Cuba, Syria, or Chechnya. For an additional 19 individuals, a history of travel or migration was reported, including New Zealand, Vietnam, India, Thailand, Greece, Iceland, Egypt, and Central and South American countries (Appendix A).

### 3.2. Antimicrobial Susceptibilities

The detailed antimicrobial resistance profiles of MRSA isolates included in our study are shown in Appendix A. All isolates were resistant to penicillin and cefoxitin, but susceptible to rifampicin, fosfomycin, linezolid, tigecycline, daptomycin, mupirocin, vancomycin, and teicoplanin. Out of 58 strains, 40 were resistant to erythromycin, 22 to ciprofloxacin and moxifloxacin, 19 to gentamicin, 10 to tetracycline, and 5 to fusidic acid (Table 1).

### 3.3. Molecular Characterization

For all isolates, the presence of the *mecA* gene was confirmed by PCR. Each strain carried the gene *lukPV*, coding for the Panton–Valentine Leukocidin.

We identified 20 different *spa* types and grouped them into four *spa* clusters using the BURP algorithm (Table 2): *spa* CC 008 (29.3%, 17/58), *spa* CC 002 (19%, 11/58), *spa* CC 019 (13.8%, 8/58), and no founder CC (17.2%, 10/58). Furthermore, there were four singletons (20.7%, 12/58). No *spa* type was excluded from analysis.

The most abundant *spa* type was t008 (22.4%, 13/58), followed by t657 (13.8%, 8/58), t002 (15.5%, 9/58), and t021 (6.9%, 4/58). The remaining *spa* types were represented by ≤ three isolates.

SCC*mec* typing revealed the presence of SCC*mec* IV (70.7%, 41/58) and SCC*mec* V (29.3%, 17/58). The majority of strains carried a SCC*mec* cassette type IV, which was subtyped into SCC*mec* IVa (*n* = 15), SCC*mec* IVb (*n* = 4), and SCC*mec* IVc (*n* = 22).

Additionally, to *lukPV*, the presence of lineage-specific CA-MRSA markers was analyzed. Seven isolates were *arcA*-positive and for five MRSA, the presence of *seh* was demonstrated. All strains were negative for *etd*.

Molecular characterization of isolates revealed the presence of various epidemic CA-MRSA lineages (Table 3). Lineages like “USA300”, including the North American Epidemic Clone (NAE) or the South American Epidemic Clone (SAE), the “Bengal Bay Clone”, and the “Sri Lankan Clone” were more prevalent than the “Taiwan Clone”, the “Southwest Pacific Clone”, or the “African Clone”.

### 3.4. Epidemiological Investigation of the Initial Cases Notified to the LHA

Thirty-nine (index) persons, diagnosed with a PVL-positive MRSA, were initially reported to the LHA in 2019. In half of these cases (21/39; 53.8%), MRSA screening of household contacts (living in the same house as the index person or having frequent contact with the index person) was performed. A total of 62 household contacts were screened, 18 (29%) of which were colonized with a PVL-positive MRSA (Table 4). In two cases, a PVL-negative MRSA was detected. For eleven index persons, where household contacts were screened, no PVL-positive MRSA was observed.

In eight cases, the contact persons were colonized with the same PVL-positive MRSA as the respective index case (Table 4). On the basis of the molecular typing results, an epidemiological linking is very likely. In the case of a large family, where a CC22-MRSA-IV was initially isolated from the abscess of the 7-year-old daughter, two other siblings were colonized with a CC8-MRSA-IV, which is different from the strain of the index case. Here, different clones seem to circulate in the family.

### 3.5. MRSA Decolonization

MRSA decolonization was performed in 25 cases; the preparations used are listed in Table 5. No data were available for the remaining 33 cases. The first decolonization cycle performed for 25 cases was successful in 9 and not successful in 4 cases (Table 6). The success of the decolonization could not be followed up in 12 cases. In four cases, a second decolonization cycle followed, which was successful for three of them; one case could not be further pursued.

## 4. Discussion

The current study provides insight into the epidemiology of PVL-positive MRSA, which were collected and analyzed from the city of Leipzig in 2019 as part of the reporting obligation in Saxony. Concrete infection control measures were carried out by the LHA, including epidemiological investigations of the initial cases to identify possible positive contact persons and to support MRSA eradication.

Since 2012, there has been a reporting obligation in Saxony for the detection of PVL-positive, community-acquired MRSA, when evidence indicates acute infection [10]. The annual report of the state authority reveals that reporting rates for the whole of Saxony have been increasing, from 20 cases reported in 2015 to 94 cases in 2021 [35]. Data on epidemiological investigations concerning those cases are not available. These are small numbers of reported cases from Saxony, but there are no data for comparison at the federal state level for Germany.

Nevertheless, there are individual studies from Germany on the characterization of PVL-positive MRSA in specific settings, e.g., in pediatric patients at a single hospital in Hesse [5], in a sentinel surveillance enrolling patients with travel-associated SSTI [36] or in isolates collected in Northern Bavaria over eight years [37].

The infections from which the strains in our study were isolated were primarily community-onset SSTIs. Furunculosis and recurrent abscesses are characteristic manifestations of PVL-positive MRSA and MSSA acquired and spread outside of medical facilities. Although the role of PVL in the pathogenesis of *S. aureus* infection is debated, PVL-producing *S. aureus* lead more frequently to recurrent skin and soft-tissue infections compared to PVL-negative *S. aureus* [11]. Furthermore, several publications correlated disease severity with the presence of the toxin [38,39].

Molecular typing revealed that the PVL-positive CA-MRSA collected in this study represent different epidemic clonal lineages, with some of them named after the regions where they were first described or where they occur frequently. Compared to other parts of the world, where single clones are dominating, the molecular epidemiology of CA-MRSA in Europe is currently characterized by heterogeneity [40]. The strains of the 39 index persons initially reported to the LHA and, of them, 19 household contacts were assigned mainly to the following (epidemic) clones: ST772-MRSA-V (“Bengal Bay Clone”), ST8-MRSA-IVa (“USA300 North American Epidemic Clone”), ST8-MRSA-IVc (“USA300 South American Epidemic Clone”), and CC5-MRSA-IVc (“Sri Lankan Clone”).

Eight isolates in our study were assigned to the “Bengal Bay Clone” (ST772-MRSA-V), which is a multidrug-resistant CA-MRSA lineage from the Indian subcontinent. It was first described in 2004 from hospitals in Bangladesh and from community settings in India. In the last decade, this clone has been isolated from MRSA cases on many continents, including Europe, the Middle East, Australia, and Africa. Apart from SSTI, it is linked to severe manifestations such as bacteremia and necrotizing pneumonia. Outbreaks in communities are described worldwide and are often associated with travel or family visits to Bangladesh, India, or neighboring countries (e.g., Nepal and Pakistan) [22]. Epidemiological investigations in our study revealed two clusters, were the “Bengal Bay Clone” circulated within the family setting (one index person suffering from SSTI and two MRSA-positive household contacts each). All persons had an Indian migration background or travel history. We characterized 15 isolates in our study as the highly virulent CA-MRSA ST8 “USA300 Clone”. These included 11 index cases, almost all of which had an SSTI, and four household contacts who were colonized with the corresponding strain. In the early 2000s, strains of this clonal lineage spread rapidly in community settings in the USA and also invaded hospitals, where they became the predominant nosocomial MRSA. In contrast, despite multiple entries, CA-MRSA “USA300” have not become permanently established in Europe either in hospitals or the community [25,41]. Seven strains were assigned to the NAE (American Epidemic) USA300 lineage, carrying *lukPV*, SCC*mec* Iva, and *arcA* as part of the ACME cluster, and eight strains showed characteristics of the SAE (South American Epidemic) USA300 lineage, including *lukPV*, SCC*mec* IVc, and the COMER element. Interestingly, six of the cases reported a history of migration or travel to Cuba and/or Mexico.

The “Sri Lankan Clone” (CC5-MRSA-IVc) was first described in 2019, being responsible for the majority of clinical CA and HA infections in a teaching hospital in Sri Lanka [33]. This clone was also identified among isolates from the United Kingdom, Australia, and the United Arab Emirates, whereas in some cases, a travel history to Sri Lanka was reported [32]. We assigned a total of 11 strains from our study to this clone, four being individual index cases and one family cluster (strains from one index person and six household contacts). Two of the individual index cases had a known history of migration or travel to Georgia and New Zealand. Within the family cluster, in addition to the index person who was hospitalized with a urinary tract infection, there were two household contacts with an SSTI and four household contacts who were colonized with the respective strain. All of them reported a history of migration or travel to Chechnya.

Data on MRSA are rarely available for some countries from which our patients migrated or have traveled (e.g., Georgia, Chechnya, Afghanistan). However, in some cases, the import of the respective strain from countries was examined in our study, where this CA-MRSA is more prevalent, is very likely (e.g., India). Thus, our data show, also like other single studies in Germany, that CA-MRSA could have been acquired in endemic areas by international travelling or migration and spread in the family environment [5,27,36,37]. In contrast to recent publications from Germany [5,13,37], we did not detect isolates of the “European CA-MRSA clone” (CC80-MRSA-IV) in our study. Originating in Sub-Saharan Africa, a strong expansion of the “European clone” occurred in the 1990ies in Europe, the Middle East, and Northern Africa [42].

Compared with PVL-negative *S. aureus*, patients colonized with a PVL-producing *S. aureus* more often suffer from recurrent episodes of SSTI [43]. This situation can have a significant impact on the quality of life and mental health of the affected individuals. Many patients go through surgery and antibiotic treatment repeatedly before colonization with PVL-positive *S. aureus* is diagnosed [11,44]. In addition to the acute therapy of the SSTI, topical decolonization is recommended to prevent recurrent infections. Furthermore, the patient should receive recommendations on hygiene measures and household contacts should be involved as early as possible, as persistent colonization of household members and contamination of environmental surfaces are highly associated with recurrent SSTI [11,38,45]. For 21 index cases in our study, the LHA was able to identify household contacts and perform MRSA screening. Thus, for half of the affected patients, MRSA-colonized household contacts could be identified. Concerning MRSA decolonization and follow-up, only limited data are available for the cases described in our study.

The major limitation in our study was patient compliance. The patient was asked by the LHA by mail to respond for a counseling interview, to take concrete infection control measures, a maximum of two times; if there was no response, no further legal action was taken. MRSA decolonization was conducted by the patient alone in the home environment. Even if the importance of compliance with decolonization treatment and repeated screening (follow-up) for successfully eradicating MRSA had been pointed out during the interviews, errors, and non-compliance could not be excluded.

The obligation to report PVL-positive MRSA enables us to detect the occurrence of PVL-producing MRSA and its spread in the population as early as possible. The cases from the city of Leipzig in 2019, which were analyzed in the present study, showed the diversity of CA-MRSA clones circulating in the community and also that household contacts are often colonized with the respective strain. For the LHA, the investigation of the reported cases was often complex, labor- and staff-intensive, and tracing back patients was challenging. As PVL-positive *S. aureus* are more frequently associated with (recurrent) SSTI, it is important to raise awareness among physicians, especially in primary care, to improve early diagnostics and treatment.

## Figures and Tables

**Figure 1 microorganisms-11-01437-f001:**
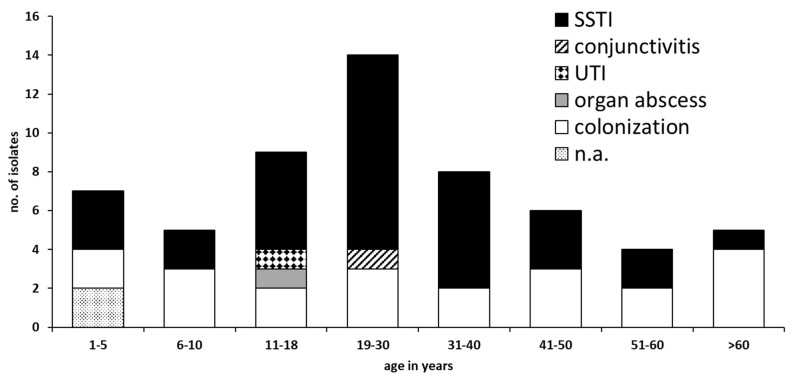
Age distribution of cases with MRSA infection or colonization. (SSTI: skin and soft-tissue infection; UTI: urinary tract infection; n.a.: data not available).

**Table 1 microorganisms-11-01437-t001:** Resistance to indicator substances of various antimicrobial classes other than ß-lactam antimicrobials. All isolates were susceptible to rifampicin, fosfomycin, linezolid, tigecycline, daptomycin, mupirocin, vancomycin, and teicoplanin.

Antimicrobial Agent	No. of Resistant Strains *n* (%)	
Ciprofloxacin	22	(37.9)
Moxifloxacin	22	(37.9)
Erythromycin	40	(69.0)
Clindamycin	3	(5.2)
Gentamicin	19	(32.8)
Tetracycline	10	(17.2)
Trimethoprim/Sulfmethoxazole	1 *	(1.7)
Fusidic acid	5	(8.6)

* I: susceptible, increased exposure (according to EUCAST).

**Table 2 microorganisms-11-01437-t002:** Characteristics of Methicillin-resistant *S. aureus* (MRSA) isolated in our study. Strains were grouped in *spa* clonal complexes (*spa* CC) according to their *spa* type. *lukPV* (Panton–Valentine Leukocidin gene), *she* (enterotoxin H gene), *arcA* (arginine deiminase gene), *etd* (gene for exfoliative toxin D), *spa* (staphylococcal protein A), SCC*mec* (staphylococcal cassette chromosome *mec*).

*Spa* CC	*n*	*spa* Type	SCC*mec* Type	CA-MRSA Marker Genes			
				*arcA*	*etd*	*lukPV*	*seh*
002	9	t002	IV c	-	-	+	-
	1	t088	IV c	-	-	+	-
	1	t105	IV c	-	-	+	-
008	7	t008	IV a	+	-	+	-
	1	t008	IV a	-	-	+	-
	5	t008	IV c	-	-	+	-
	1	t068	IV c	-	-	+	-
	2	t6172	IV c	-	-	+	-
	1	t8746	IV b	-	-	+	-
019	1	t019	IV c	-	-	+	-
	4	t021	V	-	-	+	-
	3	t363	IV a	-	-	+	-
no founder	1	t005	IV a	-	-	+	-
	1	t005	IV c	-	-	+	-
	1	t852	IV c	-	-	+	-
	2	t127	V	-	-	+	+
	3	t5388	IV b	-	-	+	+
	1	t3523	V	-	-	+	-
	1	t437	V	-	-	+	-
Singleton	1	t034	V	-	-	+	-
Singleton	8	t657	V	-	-	+	-
Singleton	2	t665	IV a	-	-	+	-
Singleton	1	t692	IV a	-	-	+	-

**Table 3 microorganisms-11-01437-t003:** Assignment of Methicillin-resistant *S. aureus* (MRSA) of our study to epidemic CA-MRSA clones. Isolates were grouped according to their *spa* type, SCC*mec* type, presence of *lukPV*, and additional characteristics associated with the respective epidemic clone. NAE (North American Epidemic Clone), SAE (South American Epidemic Clone), *lukPV* (Panton–Valentine Leukocidin gene), *seh* (enterotoxin H gene), *arcA* (arginine deiminase gene), *etd* (gene for exfoliative toxin D), COMER (copper and mercury resistance element), FUS (Fusidic acid), R (resistance), *spa* (staphylococcal protein A), SCC*mec* (staphylococcal cassette chromosome *mec*).

Strain Affiliation	Epidemic Clone	*n*	*Spa* Type(s)	SCC*mec* Type(s)	Characteristics	Patient´s History of Travel or Migration (Country (*n*))	Endemic Area (Reference)
ST772-MRSA-V	Bengal Bay clone	8	t657	V	*lukPV*+, multiresistant	India (6), Afghanistan (1)	India, Pakistan, Bangladesh [22]
CC59-MRSA-V	Taiwan clone	2	t437, t3523	V	*lukPV*+	Thailand (1)	Asia [23]
ST8-MRSA-IVa	USA300-NAE	7	t008	IVa	*arcA*+, *lukPV*+	Cuba (3), Cuba + Mexico (1)	USA [24,25]
ST8-MRSA-IVc	USA300-SAE	8	t008, t058, t6172	IVc	*lukPV*+, COMER	Cuba (2), Syria (1)	Latin America [25,26]
ST30-MRSA-IV	Southwest Pacific clone	3	t019, t665	IVa, IVc	*lukPV*+	Greece + Mauritius (1), Egypt + Dominican Republic + Mexico (1)	Southwest Pacific [27,28]
ST88-MRSA-IV	African clone	1	t692	IVa	*lukPV*+		Africa [9,29]
CC1-MRSA-V		2	t127	V	*seh*, *lukPV*+, FUS-R	Cuba (1), Syria (1)	Abu Dhabi (sporadic) [30]
CC1-MRSA-IV		3	t5388	IVb	*seh*, *lukPV*+, FUS-R	Vietnam (1), New Zealand (2)	Australia (sporadic) [30]
CC30-MRSA-V		4	t021	V	*lukPV*+	Iceland (1)	Egypt (sporadic) [31]
CC5-MRSA-IVc	Sri Lankan clone	11	t002	IVc	*lukPV*+	Georgia (1), Chechnya (7), New Zealand (1)	Sri Lanka, UAE, UK, Australia, Senegal (sporadic) [30,32,33]
CC22-MRSA-IV		3	t005, t852	IVa, IVc	*lukPV*+	Afghanistan (1), Iran (1)	worldwide [30]
CC8-MRSA-IV		5	t008, t363, t8746	IVa, IVb	*lukPV*+	Afghanistan (3), Peru + Ecuador + Bolivia + Columbia + Thailand (1)	
CC398-MRSA-V		1	t034	V	*lukPV*+		Asia [30,34]

**Table 4 microorganisms-11-01437-t004:** Patient characteristics and results of epidemiological investigations concerning those index cases, where MRSA screening of household contacts was performed and MRSA were detected. (HSP (hospital), OPD (outpatient department)).

Characteristics of Index Person		PVL-MRSA Detected		Strain Characteristics	No. of Household Contacts		Relationship to the Index Person	PVL-MRSA Strain Identical with Index
Age	Gender	Where	From		Screened	Positive for PVL-MRSA		
7	female	OPD	abscess	CC22-MRSA-IV	9	2	sibling	no
79	male	HSP	screening	ST8-MRSA-IVc	1	1	partner	yes
49	female	OPD	screening	CC1-MRSA-IV	3	2	parent/other	yes
4	male	HSP	abscess	ST772-MRSA-V	4	2	parent/other	yes
13	female	HSP	abscess	CC30-MRSA-V	7	1	parent	yes
18	female	HSP	UTI	CC5-MRSA-IVc	6	6	sibling/parent/other	yes
8	female	OPD	screening	ST772-MRSA-V	5	2	sibling/parent	yes
24	female	HSP + OPD	abscess	ST8-MRSA-IVc	4	1	partner	yes
33	female	OPD	abscess	ST8-MRSA-IVa	2	1	partner	yes

**Table 5 microorganisms-11-01437-t005:** Preparations used for MRSA decolonization (n.a., not available).

MRSA Decolonization (*n* = 25)		
Intranasal Application (*n*)	Throat Rinse (*n*)	Body Wash (*n*)
mupirocin (17)	octenidine (11)	octenidine (18)
octenidine (2)	chlorhexidine (6)	n.a. (7)
n.a. (7)	polyhexanide (1)	
	n.a. (8)	

**Table 6 microorganisms-11-01437-t006:** Success of MRSA decolonization (n.a. not available).

Decolonization Cycle	Decolonization Successful (*n*)		
	yes	no	n.a.
1.	9	4	12
2.	3	0	1

## Data Availability

Genome raw data of 19-00646, 19-00928, 19-01169, 19-01274, 19-01359, 19-02721, 19-03002 and 19-03580 were deposited to the Sequence Read Archives (SRA) (accession numbers: available at release on 31 May 2023) under BioProject number PRJNA957965.

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
