# Peer review of "Mandatory Notification of Panton–Valentine Leukocidin-Positive Methicillin-Resistant Staphylococcus aureus in Saxony, Germany: Analysis of Cases from the City of Leipzig in 2019"

_microorganisms, 2023, doi:10.3390/microorganisms11061437_

Round 1

Reviewer 1 Report

This study reported about the unique results of the mandatory reporting of PVL-positive MRSA in the state of Saxony, Germany. This study certainly provided interesting results, however, several concerns should be addressed.

1. It was understood that the law mandates the obligation of reporting PVL-positive MRSA in Saxony, but this system should be quite exceptional. Why did authors decide to report their results in an international journal? What is the point which readers outside of Germany would be interested in their study? The authors should make it clear the reason why they commenced their study, and mention it in the Introduction.

2. The method for the detection of PVL is lacking.

3. Was there any difference in the antimicrobial susceptibilities between PVL-positive and PVL-negative MRSAs?

4. The method of the decontamination of MRSA was not described in the Method section. In addition, it was mentioned that the decolonization was conducted by the patients alone in the home environment. However, application of decolonization agents by patients without the supervision of healthcare workers has the risk of suboptimal use of the agents, thus could lead to resistance to these agents. Is it a proper method for infection prevention measure?

5. It is difficult to understand the conclusion of this study. Especially what is the strength of this study? It should be described in the Discussion section.

There were several sentences which were difficult to understand. Grammatical check-up by native scholar may be necessary.

Author Response

Response to Reviewer 1:

Comment:

1. It was understood that the law mandates the obligation of reporting PVL-positive MRSA in Saxony, but this system should be quite exceptional. Why did authors decide to report their results in an international journal? What is the point which readers outside of Germany would be interested in their study? The authors should make it clear the reason why they commenced their study, and mention it in the Introduction.

Answer:

We added the following paragraph to the introduction:

Our study shows how a surveillance system for PVL-positive MRSA contributes to valuable data for molecular surveillance of the pathogen and enables early diagnosis and treatment. It should raise awareness to the related disease manifestations and targeted diagnostic evaluation especially among physicians in primary healthcare settings, in order to improve patient care.

Comment:

2. The method for the detection of PVL is lacking.

Answer:

The multiplex PCR which was used to detect markers associated with CA-MRSA, including the Panton-Valentine leucocidin gene (lukPV), is already described in line 121 ff. (2.4. Detection of resistance and virulence genes) of the M&M section.

Comment:

3. Was there any difference in the antimicrobial susceptibilities between PVL-positive and PVL-negative MRSAs?

Answer:

All MRSA described in this study are PVL positive, thus we cannot answer this question.

Comment:

4. The method of the decontamination of MRSA was not described in the Method section.

Answer:

We added the method for MRSA decolonization in the M&M section: To prevent recurrent infection, MRSA decolonization was performed, consisting of 5 days of topical therapy of the nasal mucosa, antiseptic skin cleansing and throat rinsing [11].

Comment:

In addition, it was mentioned that the decolonization was conducted by the patients alone in the home environment. However, application of decolonization agents by patients without the supervision of healthcare workers has the risk of suboptimal use of the agents, thus could lead to resistance to these agents. Is it a proper method for infection prevention measure?

Answer:

Performing decolonization in the home environment is common practice in Germany. Patients are informed in detail in advance how this is to be carried out. We have already mentioned in the limitations that the success of decolonization also depends on patient compliance: The major limitation in our study was patient compliance. …. MRSA decolonization was conducted by the patient alone in the home environment. Even if the importance of compliance with decolonization treatment and repeated screening (follow up) for successfully eradicating MRSA had been pointed out during the interviews, errors and non-compliance could not be excluded.

Comment:

5. It is difficult to understand the conclusion of this study. Especially what is the strength of this study? It should be described in the Discussion section.

Answer:

We changed the last paragraph in the discussion as follows:

The obligation to report PVL-positive MRSA enables us to detect the occurrence of PVL-producing MRSA and its spread in the population as early as possible. The cases from the city of Leipzig in 2019, which were analyzed in the present study, showed the diversity of CA-MRSA clones circulating in the community and that also household contacts are often colonized with the respective strain. For the LHA, the investigation of the reported cases was often complex, labor- and staff-intensive and tracing back patients was challenging. As PVL-positive S. aureus are more frequently associated with (recurrent) SSTI, it is important to raise awareness among physicians especially in primary care, to improve early diagnostics and treatment.

Reviewer 2 Report

Line 272-274 " The isolates analyzed in the present study represent more than half of all PVL- positive MRSA reported to the state health authority from all over Saxony in 2019  (n=71)" – it is here too not clear what are the numbers you are referencing – it would be clearer is you stated X number of cases were reported Y are isolates from family members that have been found through epidemiological study. The way it is now written it is never clear how many samples are analyzed and what is their origin.

The fact that the case of the Bengal Bay clone is described – but it is not in the results section and comes with no statistical test takes away from the article.

Line 318:  " USA300 lineage… Interestingly, six of the cases reported a history of migration or travel to Cuba and/ or Mexico" –  were there only 6 cases of this lienage in the study? It should be stated clearly here. I can see the data in table 5 – why not have it in the text? It would improve the legibility of the paragraph.

The lineages described – There should be a test to test correlation to the countries the patients emigrated from/visited over the years. Were patients with the Sri-Lankan clone originally from that part of the world or visitors on that region? You describe visits to New Zealand and Georgia and Chechnya – that does not seem to correspond – unless there is a history of Sri Lankan travel as you clearly state in lines 324-325.

Minor comments:

Line 382 "…controversially discussed" – This is awkwardly phrased

Could patients who have never been in a region where the lienage they are infected with is endemic have contacted the bacteria on hospital visits/from patients in the same room/medical staff? This might also explain the family cluster where one girl had a different type than all the rest of the family.

Slight text editing is required, but mild in nature

Author Response

Response to Reviewer 2:

Comments:

Line 272-274 " The isolates analyzed in the present study represent more than half of all PVL- positive MRSA reported to the state health authority from all over Saxony in 2019  (n=71)" – it is here too not clear what are the numbers you are referencing – it would be clearer is you stated X number of cases were reported Y are isolates from family members that have been found through epidemiological study. The way it is now written it is never clear how many samples are analyzed and what is their origin.

Answer:

The reference to which we refer is the annual report of the state authority. This only contains the number of reported cases for the whole of Saxony (n=71 in 2019), further epidemiological studies concerning those cases are not described. We have shortened and adapted the section accordingly and hope that the statements are now more understandable.

Comment:

The fact that the case of the Bengal Bay clone is described – but it is not in the results section and comes with no statistical test takes away from the article.

Aswer:

The Bengal Bay Clone is addressed several times in the results section (see also 3.3. Molecular characterization, table 3, table 4). We included the number of our cases associated with this clone in the discussion, as we did it for the other lineages. Furthermore, we shortened the section.

Comment:

Line 318:  " USA300 lineage… Interestingly, six of the cases reported a history of migration or travel to Cuba and/ or Mexico" –  were there only 6 cases of this lienage in the study? It should be stated clearly here. I can see the data in table 5 – why not have it in the text? It would improve the legibility of the paragraph.

Answer:

At the begin of the paragraph concerning USA 300 (see line 308 ff.) you will find the following sentence: “We characterized 15 isolates in our study as the highly virulent CA-MRSA ST8 “USA300 Clone”. These included 11 index cases, almost all of which had an SSTI, and four household contacts who were colonized with the corresponding strain.”

Accordingly, we do not change anything in the paragraph.

Comment:

The lineages described – There should be a test to test correlation to the countries the patients emigrated from/visited over the years. Were patients with the Sri-Lankan clone originally from that part of the world or visitors on that region? You describe visits to New Zealand and Georgia and Chechnya – that does not seem to correspond – unless there is a history of Sri Lankan travel as you clearly state in lines 324-325.

Answer:

Thank you for this comment. We are aware that the regions where specific lineages of CA-MRSA are described do not correlate with the patient's history of travel or migration. We are not claiming that these strains were acquired abroad (accordingly, we refrain from statistical analysis), it is just a possibility in some cases, what we already addressed in the discussion:

“In some cases examined in our study the import of the respective strain from countries, where this CA-MRSA is more prevalent, is very likely. Thus, our data show, also like other single studies in Germany, that CA-MRSA could have been acquired in endemic areas by international travelling or migration and spread in the family environment.”

A Sri Lankan clone may be endemic in other parts of the world where it has not yet been described. Furthermore, data on MRSA are rarely available for some countries from which our patients migrated or have traveled (e.g. Georgia, Chechnya, Afghanistan). We also added this sentence to the paragraph.    

Comment:

Line 382 "…controversially discussed" – This is awkwardly phrased

Answer:

Line 282: we changed the phrase to “debated”

Comment:

Could patients who have never been in a region where the lienage they are infected with is endemic have contacted the bacteria on hospital visits/from patients in the same room/medical staff? This might also explain the family cluster where one girl had a different type than all the rest of the family.

Answer:

Of course, that can definitely be the case. The children can have acquired the respective strain both in a medical facility or in the community (e.g. kindergarten).

Reviewer 3 Report

The authors have provided a concise description of the work and the value that such reporting provides. For the manuscript, I have no edits necessary prior to publication, except if they authors feel like they can add a translated from German to English reporting forms to the supplement.

Author Response

Response to Reviewer 3:

Comment:

The authors have provided a concise description of the work and the value that such reporting provides. For the manuscript, I have no edits necessary prior to publication, except if they authors feel like they can add a translated from German to English reporting forms to the supplement.

Answer:

We translated the two forms from German to English; available in the Supplement.

Round 2

Reviewer 1 Report

All concerns were addressed properly.

Minor grammatical errors were found. Professional editing is recommended.